# The Identification of Cu–O–C Bond in Cu/MWCNTs Hybrid Nanocomposite by XPS and NEXAFS Spectroscopy

**DOI:** 10.3390/nano11112993

**Published:** 2021-11-07

**Authors:** Danil V. Sivkov, Olga V. Petrova, Sergey V. Nekipelov, Alexander S. Vinogradov, Roman N. Skandakov, Sergey I. Isaenko, Anatoly M. Ob’edkov, Boris S. Kaverin, Ilya V. Vilkov, Roman I. Korolev, Viktor N. Sivkov

**Affiliations:** 1Federal State Budgetary Educational Institution of Higher Education, Saint-Petersburg State University, 199034 St. Petersburg, Russia; asvinograd@yahoo.de; 2Komi Science Centre of the Ural Branch of the Russian Academy of Sciences, 167982 Syktyvkar, Russia; nekipelovsv@mail.ru (S.V.N.); scanick@yandex.ru (R.N.S.); isaenko@geo.komisc.ru (S.I.I.); sivkovvn@mail.ru (V.N.S.); 3G.A. Razuvaev Institute of Organometallic Chemistry of the Russian Academy of Sciences, 603950 Nizhny Novgorod, Russia; amo@iomc.ras.ru (A.M.O.); kaverin@iomc.ras.ru (B.S.K.); mr.vilkof@yandex.ru (I.V.V.); 4Institute of Exact Sciences and Information Technologies, Pitirim Sorokin Syktyvkar State University, 167001 Syktyvkar, Russia; korolev36a@gmail.com

**Keywords:** XPS, NEXAFS, Raman shift, Cu–O–C bonding, MWCNT, Cu/MWCNTs

## Abstract

The results of the research of a composite based on multi-walled carbon nanotubes (MWCNTs) decorated with CuO/Cu_2_O/Cu nanoparticles deposited by the cupric formate pyrolysis are discussed. The study used a complementary set of methods, including scanning and transmission electron microscopy, X-ray diffractometry, Raman, and ultrasoft X-ray spectroscopy. The investigation results show the good adhesion between the copper nanoparticles coating and the MWCNT surface through the oxygen atom bridge formation between the carbon atoms of the MWCNT outer graphene layer and the oxygen atoms of CuO and Cu_2_O oxides. The formation of the Cu–O–C bond between the coating layer and the outer nanotube surface is clearly confirmed by the results of the O 1s near edge X-ray absorption fine structure (NEXAFS) and X-ray photoelectron spectroscopy (XPS) of the Cu/MWCNTs nanocomposite. The XPS measurements were performed using a laboratory spectrometer with sample charge compensation, and the NEXAFS studies were carried out using the synchrotron radiation of the Russian–German dipole beamline at BESSY-II (Berlin, Germany) and the NanoPES station at the Kurchatov Center for Synchrotron Radiation and Nanotechnology (Moscow, Russia).

## 1. Introduction

The discovery of multi-walled carbon nanotubes (MWCNTs) in 1991 [1] and the development of technologies for their production in macroscopic quantities [2,3,4,5,6,7] laid the foundation for systematic research aimed at solving the problem of the practical application of MWCNTs. Due to their high mechanical strength, large specific surface area, and chemical inertness, carbon nanotubes can be effectively used as a base of catalysts in chemical technology. The use of MWCNTs allow stabilizing the deposited nanoparticles of metals and their carbides or oxides, thereby increasing their high specific surface area and significantly changing their physicochemical properties [8]. Among the variety of methods for obtaining MWCNT-based nanocomposite materials (electrochemical reduction of metal salts via a sol-gel process, metal organic chemical vapor deposition (MOCVD), electrolysis, and physical deposition, such as electron beam spraying, thermal spraying, etc.), the MOCVD method, which consists of the deposition of metal-containing nanoparticles to the MWCNTs surface during the pyrolysis of metal organic compound (MOC) vapors as a result of chemical reactions on the MWCNTs surface, has a number of advantages. Due to a variety of organometallic compounds, relatively low pyrolysis temperatures, and easy removal of volatile reaction products from the solid-state phase deposition zone, the method makes it possible to regulate the coating composition and the rate of the synthesis process. MOCVD was successfully used to form metal-containing nanostructured coatings of pyrolytic iron [9,10], chromium [11,12], titanium carbide [13], tungsten carbide [14], rhenium [15], aluminum [16], and bimetallic rhenium–tungsten nanodendrites (Re-W/MWCNTs) [17].

Recently, in [18], an inorganic compound, copper formate (a salt of copper and formic acid), was used as a precursor for the copper deposition. The Cu/MWCNTs nanocomposite was produced by pyrolysis in an argon flow of copper formate previously deposited on the MWCNT surface from an aqueous solution. The obtained material was successfully used as a catalyst in a “chloride technology” application for obtaining high-purity monocrystalline germanium. The nanocomposite usage allows achieving important practical results: reducing the reaction temperature and obtaining a germanium tetrachloride conversion of 98%. The developed technology of copper nanoparticle deposition on the MWCNT surface also demonstrated its effectiveness for the composite material preparation. However, a number of problems, associated with elucidating the mechanism of the interaction between the MWCNTs surface and the copper coating, the chemical state of carbon atoms on the outer wall of the MWCNTs, the structure and chemical composition of the MWCNTs-metal interface and coating layer, remain unresolved. Understanding the nature of the adhesion between the metal coating and the outer graphene layer of the MWCNTs is an urgent problem, and its solution is important for the application of metal/MWCNTs nanocomposites. “Metallic” surfaces, in most cases, are a mixture of metal oxides, and, therefore, the crucial question is: do the carbon atoms of the graphene layer bind directly to the metal atoms, forming a metal–carbon bond, and/or through an oxygen atom, forming a metal–oxygen–carbon bond? In our previous study [12] with using the near edge X-ray absorption fine structure (NEXAFS) and the X-ray photoelectron (XPS) spectroscopy, it was revealed that, in Cr/MWCNTs and Fe/MWCNTs nanocomposites, metal–carbon bonds (Cr–C and Fe–C, respectively) are realized. In the works [19,20], the formation of oxygen bridges between the graphene and NiO nanosheets [21], and Fe_3_O_4_ nanoparticles and graphene [19], was shown using the O 1s XPS spectra.

In turn, the O 1s spectra of CuO/graphene nanosheets (CuO/GNSs) [20] indicate the good adhesion between the copper-containing coatings and oxidized graphene layers due to the Cu–O–C bonding. However, in this work, the quantitative assessment of the Cu–O–C contributions to the O 1s XPS spectra was complicated by the superposition of structures associated with graphene and copper oxides since their concentrations were comparable in CuO/GNSs samples. Considering the above, the main aim of this work is to obtain and conduct comprehensive studies of Cu/MWCNTs nanocomposites with a low concentration of carbon and copper oxides using copper formate thermal decomposition. This will enable the unambiguous identification of the structural elements corresponding to Cu-O-C bonding in O 1s XPS and NEXAFS spectra.

## 2. Materials and Methods

### 2.1. Materials

The initial MWCNTs and Cu/MWCNTs composite were synthesized in the Hybride Nanomaterials Laboratory of G.A. Razuvaev Institute of Organometallic Chemistry of the Russian Academy of Sciences (RAS).

Materials list: especially pure toluene (produced by JSC ECOS, Moscow, Russia); ferrocene (C_10_H_10_)Fe (98%, produced by Aldrich, St. Louis, MO, USA); copper formate (H-COO)_2_Cu (97%, produced by Aldrich, St. Louis, MO, USA); gaseous argon (produced by LLC SPE “Salyut-gas”, Nizhny Novgorod, Russia, TU 2114-011-106-818-63-2005). The volume fraction of argon is not less than 99.994%.

For the ultrasoft X-ray spectroscopy, the oxides of copper (I) and copper (II) of special purity grade (produced by LLC UPCP, Verkhnaya Pyshma, Russia) were used.

### 2.2. Synthesis

For the synthesis of MWCNTs, the MOCVD method via the pyrolysis of ferrocene and toluene mixture in an argon flow (flow rate 500 cm^3^/min) at atmospheric pressure, a tubular quartz reactor was used. The synthesis procedure is described in detail elsewhere [22]. The schematic illustration of the experimental setup used for the synthesis of radial-oriented aligned MWCNTs is presented in Appendix A.

The Cu-containing nanoparticles decoration of the MWCNT surface was carried out in a pyrex glass ampoule with a side outlet at 473 K in a high-purity argon flow. The experimental setup used for the Cu/MWCNTs composite synthesis is presented in Appendix A. The deposition of the nanoparticles onto the MWCNT surface upon the copper formate pyrolysis with the formation of a Cu/MWCNTs nanocomposite occurred according to the scheme
(H-COO)_2_Cu → Cu + 2CO_2_ + H_2_.(1)

After being taken out of the reactor, the Cu/MWCNTs nanocomposite did not required further treatment in any solvents and was ready for further application. The nanocomposite was stored in a volume filled with high-purity argon.

### 2.3. Characterization

The initial MWCNTs and Cu/MWCNTs nanocomposite were characterized using a complementary set of chemical and physical analysis methods: transmission, high-resolution transmission and scanning electron microscopy (TEM, HRTEM, and SEM), X-ray diffraction (XRD), and energy-dispersive X-ray (EDS), Raman, XPS, and NEXAFS spectroscopy.

#### 2.3.1. XRD, TEM, SEM, and EDS

XRD analysis of the MWCNT samples and hybrid materials was performed with CuK_α_-radiation using Bruker D8 Discover X-ray diffractometer (Bruker Corporation, Billerica, MA, USA) in the θ-2θ symmetrical geometry with a Gobel mirror, an equatorial Soller slit with the angular divergence of 2.5°, and a slit of 1.5 mm on the primary beam. The obtained diffraction patterns were processed using DIFFRAC.EVA software with the PDF-2 (2012) powder diffraction database.

The surface morphology of the synthesized materials was examined by SEM using Supra 50VP (Carl Zeiss AG, Oberkochen, Germany) scanning electron microscope. The structure of the MWCNT and Cu coatings deposited on the surface of the MWCNT was studied by TEM with a LIBRA 200 MC Schottky field-emission gun instrument (Carl Zeiss AG, Oberkochen, Germany) operating at 200 kV and the information resolution limit of 0.12 nm. EDS data were obtained using a Hitachi Regulus 8100 scanning electron microscope equipped with a Bruker XFlash 6–60 energy dispersive X-ray detector.

XRD, TEM, HRTEM, SEM, and EDS measurements were conducted using equipment of the Center “Physics and technology of micro- and nanostructures” at the Institute for Physics of Microstructures of RAS (N. Novgorod, Russia) and Syktyvkar State University (Syktyvkar, Russia).

#### 2.3.2. Raman Spectroscopy

Raman spectroscopy measurements were conducted at the Centre of Collective Use at the Institute of Geology of Komi Science Centre of the Ural Branch of RAS (Syktyvkar, Russia) using a Horiba-Yvon Jobin LabRam HR800 spectrometer in air at room temperature and an Ar laser with the 1 mW power and 488 nm wavelength. Samples were analyzed with 50× and 100× lenses. To limit the power of the laser radiation, neutral filters were used. Spectra were recorded in the 100–4000 cm^−1^ range using a diffraction grating with a density of 600 lines/mm. The spectral and spatial resolutions of the spectrometer were about 1 cm^−1^ and 1 μm, respectively. Each spectrum was obtained as a result of three accumulations of 10 s each. The spectra were recorded at room temperature. After background correction, the spectrum was deconvoluted into individual lines using a curve-fitting procedure from the software provided by LabSpec 5.36.

#### 2.3.3. NEXAFS Spectroscopy

NEXAFS spectra of the initial MWCNTs, the Cu/MWCNTs nanocomposite, and the powder from the reactor walls, as well as the reference compounds (metallic copper, CuO, and Cu_2_O) were obtained at the Russian–German beamline (RGBL) of the Berliner Elektronenspeicherring für Synchrotronstrahlung (BESSY-II, Berlin, Germany) and NanoPES station at the Kurchatov Center for Synchrotron Radiation and Nanotechnology (KISI-Kurchatov, Moscow, Russia). The RGBL dipole beamline of BESSY-II to be highly suited for spectroscopic investigations in region of the 100–1400 eV [23,24]. All NEXAFS spectra were measured in total electron yield (TEY) mode under an ultrahigh vacuum 10^−9^ mbar at room temperature. Energy calibration was done by the energy separation between the first- and second-order light-excited Au 4f_7/2_ photoemission lines measured from a clean gold plate fixed together with the sample on the holder. In the case of C 1s NEXAFS spectra, the energy calibrations were performed using the well-resolved π*-resonance at 285.38 eV in the HOPG C 1s spectrum [25]. The photon flux spectral dependence was determined using a clean Au photocathode, the degree of surface cleanliness of which had been preliminarily checked by X-ray photoelectron spectroscopy. The photon flux was determined from the TEY curve of the Au plate dividing by the well-known atomic X-ray absorption cross section of Au [26] in accordance with the method described previously in [12,27]. The photon energy resolution was below 0.05 eV (C 1s edge), 0.1 eV (O 1s edge), and 0.2 eV (Cu 2p edge). The samples of the initial MWCNTs, Cu/MWCNTs, powder, CuO, and Cu_2_O oxides for absorption measurements were prepared ex situ in air by pressing of investigated material powders into the clean surface of copper, molybdenum, or indium plate. The metallic copper samples were obtained by thermal deposition of thick copper layers on the silicon plate.

#### 2.3.4. XPS

XPS studies were carried out at the resource center “Physical methods of surface investigation” of the Science Park of St. Petersburg State University (St. Petersburg, Russia). XPS analysis was performed using X-ray Photoelectron spectrometer Thermo Scientific ESCALAB 250Xi with AlK_α_ radiation (1486.6 eV) of X-ray tube to excite photoelectron (PE) spectra. The survey spectra and high-resolution core-level spectra were measured at pass energy of 100 eV and 50 eV, respectively. To neutralize the sample charging during the experiments, an electron-ion charge compensation system was used. The studies were carried out under an ultrahigh vacuum 10^−10^ mbar at room temperature; the argon partial pressure in the analytical chamber was about 2 × 10^−7^ mbar. The experimental data were processed using the ESCALAB 250 Xi spectrometer software.

## 3. Results and Discussion

### 3.1. Initial MWCNT Research

Due to electron microscopic investigation, an exhaustive description of the morphology and structure of both the initial MWCNTs and Cu/MWCNTs nanocomposite in the form of (CuO/Cu_2_O/Cu)/MWCNTs hybrid nanomaterial were obtained. Figure 1 shows the cross-sectional SEM (a), TEM (b), and HRTEM (c) images of the initial MWCNTs after grinding in a rotary-type disperser. The SEM studies revealed the presence of MWCNTs with different diameters (Figure 1a). According to the TEM data (Figure 1b), the average outer diameter of the synthesized carbon nanotubes is about 80 nm, and their lengths range from hundreds of micrometers to several millimeters. The internal channels of MWCNTs contain a small amount of catalytic iron nanoparticles, which were shown by TEM. According to the HRTEM data, the interplanar distance between the graphene planes in the MWCNTs is 0.34 nm (Figure 1c). The insert in Figure 1c shows the fast Fourier transformation (FFT) pattern obtained using Fourier filtering to weaken the background contribution. Registered FFT reflexes of graphite in the (002) direction are typical for MWCNTs.

The EDS spectrum in Figure 2a shows the dominant carbon content, a negligible amount of oxygen, and iron traces in the nanotube. In this case, Fe atoms are inside the nanotube since the XPS method demonstrates the absence of iron on the MWCNT surface (insert in Figure 2a). It follows from the XPS measurements that the relative atomic concentration of oxygen on the MWCNT surface is approximately 2%, and, according to the EDS data, 3%. The values obtained by the EDS are higher than the XPS results, which is due to the presence of a small amount of iron oxides in the MWCNT volume.

Figure 2b demonstrates the spectrum of the Raman scattering from the MWCNTs. The Raman spectrum of the initial MWCNTs exhibits two main peaks located at 1361 and 1586 cm^−1^, which have been assigned to the D-band and G-band of the MWCNT, respectively. The D band has been associated with the presence of disordered graphite domains and structural defects or impurities. The G band represents the E2g stretching vibration of the carbon atoms in a hexagonal crystal structure [28]. The intensity ratio *I_D_/I_G_* is sensitive to the defectiveness degree of the MWCNT structure. Figure 2b shows a low ratio (*I_D_/I_G_*) for the initial MWCNTs (0.45), indicating a high quality of the MWCNTs.

The O 1s and C 1s XPS spectra of the initial MWCNTs are shown in Figure 2c. The O 1s spectrum was fitted by two peaks with approximately equal intensities with binding energies (BEs) of 531.7 and 533.5 eV, assigned to the 1s core level of the oxygen atom in the C=O and C–O functional groups, respectively [29]. These peaks have approximately equal areas, which indicates that the amount of these oxygen-containing groups is approximately the same and is about 1%.

In turn, the C 1s XPS spectrum of the initial MWCNTs is composed of four C 1s PE bands. The first band A at BE of 284.5 eV corresponds to carbon atoms in graphite, the next band B (286.4 eV) reflects the presence of carbon atoms singly bonded to oxygen atoms (C–O) in MWCNT, and the third band C at BE of 288.6 eV represents the PE signal from carbon atoms doubly bonded to oxygen atoms (C=O). Finally, the last band at BE of 290.8 eV is a π-plasmon satellite of the C 1s PE line in aromatic compounds.

The C 1s NEXAFS spectra of carbon–oxygen compounds are sensitive to the presence of single (C–O), epoxy (C–O–C), and double (C=O) bonds between carbon and oxygen atoms and can be applied to the complementary analysis of the initial MWCNTs. The C 1s NEXAFS spectrum of the initial MWCNTs shown in Figure 2d is in good agreement with the data of other studies [30,31,32,33]. In addition to the π* and σ* absorption peaks characteristic of the graphene spectrum, it exhibits additional structures in the photon energy range of 287–290 eV, in which the π*-bands of the C 1s NEXAFS spectra of oxygen–carbon functional groups are usually located [30,31,32]. In Figure 2d, these peaks associated with the transitions of the C 1s core electrons to π*-unoccupied orbital of the C–O (286.4 eV), C–O–C (287.2 eV), and C=O (288.4 eV) functional groups [27,33,34] are indicated by arrows. Since the content of the carbon atoms of these groups in the MWCNTs is less than 2%, their contribution to the C 1s NEXAFS spectrum of the initial MWCNTs appears as a broad band of low intensity.

### 3.2. XRD, SEM, HRTEM, EDX, and Raman Characterization of Cu/MWCNTs

The results of the XRD analysis of the initial MWCNTs (black line) and Cu/MWCNTs nanocomposite (blue line) are shown in Figure 3. The XRD pattern of the MWCNTs correlates well with that of graphite with slightly increased interplanar spacings. The XRD phase analysis of the Cu/MWCNTs nanomaterial indicates the presence of the several main crystalline phases: the MWCNTs, the fcc copper Cu (ICDD PDF2 #00-004-0836), the monoclinic copper (II) oxide CuO (ICDD PDF2 #00-045-0937), and the cubic copper (I) oxide Cu_2_O (ICDD PDF2 #00-005-0667). It is clearly seen that the dominant phases are metallic copper, copper (I), and copper (II) oxides. The presence of these phases in the samples of synthesized nanomaterials means that they are (CuO/Cu_2_O/Cu)/MWCNTs hybrid nanocomposites. All the diffraction reflections are very broadened, which indicates the small size of the crystallites.

Cu/MWCNTs hybrid nanocomposites, as well as the initial MWCNTs, were studied using electron microscopy (Figure 4 and Appendix A). It was found that the hybrid nanomaterial is the MWCNTs decorated with composite spherical nanoparticles (Figure 4a–c). The diameter of the nanoparticles ranges from 5 to 150 nm, although the overwhelming majority of them have a size of about ~17 nm and ~45 nm. The distribution of the MWCNT and nanoparticle diameters are shown in the Appendix A, respectively. The SEM images illustrating the distribution of large nanoparticles is shown in Figure 4a or Figure 5a, and, for particles of a smaller diameter, the corresponding SEM images are given in the Appendix A. The CuO/Cu_2_O/Cu nanoparticle has a Cu crystal core enclosed in a thin polycrystalline shell, which contains CuO and Cu_2_O nanocrystallites. The thickness of the oxide shell is about 5 nm, and the sizes of the copper oxide nanocrystallites are approximately of 2–3 nm.

In the HRTEM image (Figure 4c), the system of the atomic planes of the Cu, Cu_2_O, and CuO nanoclusters in the (111) direction are shown as blue, green, and red, respectively. The FFT pattern is given in the insert in the right upper corner of Figure 4c. The FFT reflections are marked with different colors for comparison with the phase contrast data of the interatomic planes.

According to the EDS spectrum (Figure 5), the sample of hybrid material only consists of C, O, and Cu atoms. The element distribution maps demonstrate the expected presence of carbon atoms in the MWCNTs and their complete absence in the nanoparticles. In addition, a small amount of adsorbed oxygen atoms was also found on the nanotubes. Thus, only Cu and O atoms determine the chemical composition of nanoparticles in a Cu/MWCNTs hybrid nanocomposite. In other words, the obtained results suggest that this nanocomposite is an array of MWCNTs decorated by CuO/Cu_2_O/Cu heterophase nanoparticles with a copper metal core and a thin shell of a mixture of CuO–Cu_2_O oxides.

Figure 6 shows the Raman spectra of the initial MWCNTs, the Cu/MWCNTs hybrid nanocomposite, and the substance from the reactor walls (powder) formed during the copper formate thermal decomposition in the process of the nanocomposite synthesis.

The Raman spectra of the initial MWCNTs and Cu/MWCNTs hybrid nanocomposite show two major peaks, located at 1361 and at 1586 cm^−1^, which were assigned to the D-band and to the G-band of the MWCNTs, respectively. The D-band was associated with the presence of disordered graphitic domains and structural imperfections or impurities [28]. The integral intensity ratio *I_D_*/*I_G_* is close to 0.5. This suggests that the structure of the nanotubes is not destroyed during the deposition of copper. It can be seen from Figure 6a that the peaks D and G in the spectrum of the nanocomposite are broadened, probably due to the attachment of the functional groups and copper particles to the nanotube surface, which can lead to structural deformations. Furthermore, only five additional peaks are observed in the Raman spectra of the nanocomposite; three peaks at 214, 414, and 645 cm^−1^ are ascribed to Cu_2_O, and two peaks centered at 295 cm^−1^ and 628 cm^−1^ are believed to be CuO-related peaks [35,36,37,38]. This finding allows us to conclude that both CuO and Cu_2_O are present in the nanocomposite sample.

It is known that the intensity of the vibrational mode is directly proportional to the number of scattering centers present in the volume illuminated by the laser beam, and its energy depends on the distance, mass, and geometric arrangement of the atoms. The position of the Raman mode reveals indirect information on stoichiometry, while the width of the modes is linked to disorder. It can be seen from the Figure 6a that the peaks observed in the nanocomposite spectrum have a lower intensity and larger width than the ones in the powder spectrum. The low intensity and large width of the Raman peaks in the spectrum of the Cu/MWCNTs hybrid nanocomposite are due to the low concentration of copper oxides and the nanoscale size of their particles.

Figure 6b shows a review of the powder Raman spectrum. The inset shows the features associated with carbon and its oxides. In the presented spectrum, there are no structures of C–C, C=C, and C=O groups (~1500–2000 cm^−1^). The weak peaks indicate the presence of CO_3_ groups (~1080 cm^−1^) [39], C–O (~1150 cm^−1^), and C–O–C (~1280 cm^−1^) bonds [40]. In addition to the features described above, there is another peak at 800 cm^−1^. This peak is supposed to be associated with the presence of copper hydroxide, namely the Cu–OH bending modes [37,41]. The analysis of the powder Raman shift spectrum shows that, in the process of the copper formate, thermal decomposition, and its subsequent removal into the air, an insignificant amount of carbon oxides, in the form of C–O, C–O–C, and CO_3_ groups, and copper hydroxides are formed.

### 3.3. NEXAFS and XPS Study of Cu/MWCNTs

The C 1s NEXAFS spectra of the initial MWCNTs and the Cu/MWCNTs hybrid nanocomposite were obtained using the technique described earlier in [12] and successfully approved for nanocomposites based on MWCNTs coated with chromium and iron [12], and tungsten carbide [14]. Figure 7 shows the C 1s NEXAFS partial spectra of the MWCNTs and Cu/MWCNTs derived from the above spectra by subtracting the contributions from the overlying shells, obtained by the power law extrapolation of the absorption cross section dependence in the long-wave region before the C 1s absorption edge. A comparative analysis of the spectra shows that the shape and energy positions of all the features in the C 1s NEXAFS spectrum of the MWCNTs are retained in that of the Cu/MWCNTs hybrid nanocomposite. At the same time, there are significant differences in the spectrum of the nanocomposite: a 2.1-fold decrease in the area under the spectral curve, the appearance of an additional structure with three low-intensity peaks, A, B, C, and a shoulder D in the region between the π* and σ* resonances. The energy positions of the additional peaks are in good agreement with the energies of the C1s → π* transitions in the C–O (A, 286.9 eV), C–O–C (B, 287.7 eV), C=O (C, 288.6 eV, as part of the carboxy and carbonyl group), and CO_3_ (D, 290.4 eV) groups, which is confirmed by the numerous studies of carbon oxides [12,14,33,42] and various oxygen–carbon containing molecular groups [34,43,44,45,46]. However, in [47], it is noted that the C 1s → π* electron transition in the carboxy–COOH group C=O coupling lies in the 290.0–290.4 eV range, although other authors associate this transition with the absorption band at a photon energy of 288.6 eV [34,42,44,45]. It is also suggested [20] that the shoulder D at 290.4 eV in the C 1s NEXAFS spectrum of CuO/graphene reflects the presence of the strong bonding between the Cu_2_O and graphene sheet by forming a Cu–O–C bond. In our previous studies [12,14], the D band was also observed in the C 1s NEXAFS spectra of the Cr/MWCNTs, Fe/MWCNTs, and WC/MWCNTs nanocomposites. Therefore, the appearance of the D band should be associated with the formation of compounds containing anion CO_3_^2−^ during the nanocomposite synthesis.

The C 1s NEXAFS spectrum measured in the TEY mode is mainly determined by the yield of Auger and secondary electrons from the sample. Therefore, the reason for the decrease in the total oscillator strength for the C 1s NEXAFS spectrum of the hybrid nanocomposite is the lowering of the TEY signal from the MWCNTs surface due to the weakening of the flux of Auger and secondary electrons by copper-containing nanoparticles on the MWCNT surface. According to the electron microscopy and X-ray diffractometry, the particle size is 50–150 nm, which is comparable to the electron escape depth, including secondary ones (1–100 nm), from solids [48]. Therefore, the C 1s NEXAFS spectrum is the superposition of those of the nanotube open surface, interface between the copper nanoparticle, and the nanotube outer surface, and the spectra of the surface of large particles with dimensions exceeding the escape depth of X-ray photoelectrons. The ratio of the oscillator strength sum for the partial spectra of the initial nanotube and the Cu/MWCNTs hybrid nanocomposite (Figure 7) is 1/2.1 = 0.48. Consequently, large nanoparticles cover less than half of the nanotube outer surface. In the process of copper formate decomposition, the formation of carbon–oxygen, carbon–oxygen–copper, carbon–copper bonds, and copper bicarbonates are probable. The CO_3_^2−^ anion presence in the sample can be determined by the characteristic peak at 290.4 eV in the NEXAFS C 1s absorption spectrum [43,49,50]. However, the peaks specific for the carbon oxides and carbides of 3d metals located in the 286–289 eV energy range [51] cannot be identified separately. In this case, the C 1s PE spectra are more informative.

Figure 8 shows the survey and C 1s XPS spectra of the initial MWCNTs, nanocomposite, and a chemical compound formed on the walls of the reactor. The survey spectra contain O 1s and Cu 2p PE peaks, the intensity of which is several times lower in the spectrum of the nanocomposite compared to that of the powder due to a significantly lower concentration of copper compounds on the nanotube surface. The comparison between the C 1s XPS spectra of the nanocomposite (Figure 8b) and MWCNT (Figure 2c) demonstrates (i) a 1.4-fold decrease in the intensity of the main C 1s peak at BE of 284.6 eV in the spectrum of the Cu/MWCNTs hybrid nanocomposite compared to that of the initial nanotube, as well as (ii) a very low intensity of this peak in the powder spectrum. The first is caused by the attenuation of the TEY signal from the MWCNT surface by the copper nanoparticles, and the second by a small amount of carbon compounds in the powder.

A detailed examination of the C 1s XPS spectrum (Figure 8b) reveals additional bands corresponding to the ionization of the 1s core level of the carbon atom in different carbon–oxygen functional groups (C–O, C–O–C, and C=O). Additional information on the decomposition of the C 1s XPS spectra of the Cu/MWCNTs nanocomposite is presented in the Appendix A. The energy positions of the corresponding peaks are indicated by arrows in Figure 8b according to the data from the work [33]. The presence of the broad band at the BE of 289 eV in the C 1s XPS spectrum of the nanocomposite and the powder, which is absent in that of the initial MWCNTs, is apparently associated with the formation, during the nanocomposite synthesis, of a carbon compound with a C atom coordinating two or more oxygen atoms. Copper bicarbonate can act as such a compound. Its formation is possible in the course of the decomposition of the copper formate and the subsequent removal of the nanocomposite into the air where metallic copper can interact with the oxygen, carbon dioxide, and water molecules adsorbed on the surfaces according to the scheme
2Cu + CO_2_ + O_2_ + H_2_O→(CuOH)_2_CO_3_.(2)

It should be noted that the C 1s BE for the CO_3_^2−^ anion in copper bicarbonates (289.1 eV) [52] coincides with the energy position of the above broad band in the spectra of the composite and powder. This finding confirms the identification of the absorption band D in the nanocomposite C 1s NEXAFS spectrum (Figure 7) as the C 1s → π* electron transition in the group CO_3_. The lack of the PE structures characteristic of 3d-transition metal carbides [12,53] in the BE range of 283–284 eV of the C 1s XPS spectrum of the nanocomposite (Figure 8b) indicates the absence of copper carbide in the Cu/MWCNTs hybrid nanocomposite coating. This, in turn, allows to associate the A–D bands in the C 1s NEXAFS spectrum of the Cu/MWCNTs exclusively with carbon oxides. Of particular interest is the identification of the structural elements of the NEXAFS and XPS spectra, reflecting the chemical bonding between carbon and copper atoms through the oxygen atom (Cu–O–C). Presumably, this bonding provides adhesion between the copper containing nanoparticles and the MWCNT surface. However, the measured C 1s XPS and NEXAFS spectra from the Cu/MWCNTs do not make it possible to unambiguously establish the presence of this bonding in the hybrid nanocomposite.

In Figure 9a, the Cu 2p NEXAFS spectrum from the Cu/MWCNTs is compared with those of the atomic copper [54], metal copper film, powder from the reactor walls, copper oxide CuO, and cuprous oxide Cu_2_O. The fine structure of the 2p absorption spectra is determined by the dipole-allowed transitions of the core 2p electrons to the unoccupied 3d and 4s states as a result of the absorption of X-ray photons. The 3d electron shells of the metal atoms of the first transition period are spatially localized more strongly in comparison with 4s shells due to the collapse of the 3d shell [55]. In turn, the strong localization of the 3d electron shell leads to higher absorption cross sections for 2p_3/2,1/2_ → 3d transitions: in the copper atom, they are about two orders of magnitude higher than the cross sections for 2p → 4s transitions [55]. Thus, the fine structure of the Cu 2p absorption spectra turns out to be very sensitive to the degree of filling of the 3d states of Cu-atoms in various compounds.

In free copper atoms with the [Ar] 3d^10^ 4s^1^ electronic configuration, the 3d electron shell is completely filled, and, therefore, in the 2p absorption spectrum of free copper atoms, there is no structure corresponding to 2p electron transitions to 3d states, and only a low-intensity peak Cu 2p_3/2_ → 4s at the photon energy of 931.17 eV is observed [55]. Copper atoms in the metal have the electronic configuration [Ar] 3d^10−x^4s^1+x^, x = 0.4–0.45 [58,59], indicating the hybridization of the 3d and 4s electronic states of the copper atom in the valence band of the metal. As a result, the unoccupied electronic states at the bottom of the conduction band of copper also have a hybridized character with 3d electron contributions. In the Cu 2p spectrum of metallic copper, these 3d electron contributions provide a noticeable intensity of absorption 2p electron transitions to low-energy unoccupied electron states. In particular, the first absorption band at a photon energy of 933.5 eV is associated with such transitions. 

The consideration of the spectra in Figure 9a reveals that the Cu 2p_3/2_ NEXAFS spectra of the copper oxides are dominated by the intense absorption bands (CuO) and (Cu_2_O). The presence of a low-intensity peak at a photon energy of 931.3 eV in the Cu_2_O spectrum is related to the presence of the CuO admixture. The Cu 2p_3/2_ electron BE of 932.18 eV in the Cu_2_O is close to the metal BE of 932.63 eV, and, in CuO, it shifts to the high energy region 933.76 eV [56,57,60]. It should be noted that the absorption edge (the energy at half the first absorption band intensity) of the Cu_2_O sample is approximately in the same position as the one of pure Cu. However, the CuO absorption edge is shifted by about 2 eV in to the low energy region, and the first peak intensity increases more than twice. These sharp changes in the Cu 2p_3/2_ NEXAFS spectra of CuO are due to the 2p → 3d transition and the Coulomb interaction between the strongly localized 2p hole and the 2p electron excited into a rather narrow band of localized Cu 3d states. The final state is a so-called “core exciton” but not a density of state feature, which is proved by the XPS measurements: the observed peak in the NEXAFS spectrum is below the XPS derived binding energy for the oxide CuO [61].

A comparison of the Cu 2p NEXAFS spectra of metallic copper, CuO, and Cu_2_O with a low admixture of CuO (Figure 9a) shows that the Cu 2p spectra make it possible to identify the copper oxides presence in the sample by the 2p_3/2_ → 3d absorption band energy position. However, metallic copper cannot be identified due to the overlap of the Cu 2p NEXAFS spectra of Cu_2_O and Cu-metal. The comparative analysis of the Cu 2p NEXAFS spectra shows the presence of CuO and Cu_2_O oxides on the nanocomposite surface and in the powder, which is well correlated with the X-ray diffractometry, TEM, and HRTEM data. The Cu 2p NEXAFS and XPS spectra show that the contribution of CuO is much larger than that of Cu_2_O. However, this does not contradict the fact that the nanocomposite XRD pattern (Figure 3) demonstrates that Cu oxides are mainly composed of Cu_2_O. It should be taken into account that XRD is a volumetric method, while NEXAFS and XPS characterize the composition on the sample surface. Thus, taking into account that the 2p_3/2_ → 3d transition peak intensity in the Cu 2p NEXAFS spectrum of CuO is more than two times higher than that of Cu_2_O, from the ratios of the absorption bands at 931.3 eV and 933.9 eV (1:1 and 7:1 for powder and Cu/MWCNT, respectively), it can be seen that the CuO relative content in the nanocomposite is an order of magnitude higher than in the powder. A similar conclusion follows from a comparison of the Cu 2p XPS spectra of nanocomposite, powder, CuO, and Cu_2_O [57] (Figure 9b). However, the metallic copper cannot be identified by the Cu 2p XPS spectra due to the overlap of the closely located Cu 2p_3/2_ peaks of Cu_2_O and Cu-metal [60].

The presence of copper, its oxides, and bicarbonate in the samples allows to identify the peaks M (932.6 eV), M1 (932.3 eV), M2 (934.2 eV), and M3 (934.6 eV) in the Cu 2p XPS spectra of the Cu/MWCNTs and the powder as corresponding to Cu-metal, Cu_2_O, CuO, and copper bicarbonate, respectively. Moreover, the ratio of the areas under the corresponding peaks S(M):S(M_1_):S(M_2_):S(M_3_) = 6:8:76:10 shows the percentage of copper compounds in Cu/MWCNTs. Thereby, the main copper compound in the nanocomposite is CuO (76%). The Cu_2_O content (8%) is an order of magnitude less than that of CuO, which is consistent with the Cu 2p NEXAFS spectroscopy data. For the powder Cu 2p XPS spectrum decomposition, the ratio of the areas under the M, M_1_, M_2_, and M_3_ peaks is equal to S(M):S(M1):S(M2):S(M3) ≈ 21:35:30:5. Thus, the powder contains an almost equal amount of CuO and Cu_2_O oxides and 22% of metallic copper.

It is known [62] that, after removing a sample to the atmosphere, three main stages of copper oxide growth occur simultaneously and interdependently: (i) the formation of the Cu_2_O layer, (ii) the formation of a metastable upper Cu(OH)_2_ layer, and (iii) the transformation of the metastable Cu(OH)_2_ phase into a more stable CuO layer. Since the copper surface always has a layered CuO/Cu_2_O/Cu metal structure, the NEXAFS and XPS signals from the underlying layers will be attenuated by the upper ones. Therefore, the data on the content of the metallic copper and Cu_2_O in the nanocomposite and powder samples will be underestimated. An analysis of the Cu 2p NEXAFS and XPS spectra shows that, due to the superposition of the copper oxide spectra, it is impossible to distinguish the features associated with the C–O–Cu bond, as in the case of the C 1s spectra. However, since copper is divalent in the Cu–O–C bond [63] and the formation of this bond is possible during the copper deposition on the MWCNT surface, the sharp difference in the relative contents of CuO and Cu_2_O oxides in the powder and in the Cu/MWCNTs hybrid nanocomposite may be related to the formation of such a bond.

Figure 10 shows the measured O 1s NEXAFS and XPS spectra of the initial nanotube, Cu/MWCNTs hybrid nanocomposite, powder, and CuO, as well as Cu_2_O spectra taken from elsewhere [57,61]. These NEXAFS spectra were normalized to the TEY signal of the Au mesh with the 50% transmission (the initial TEY signal given in the Appendix A). The O 1s NEXAFS spectrum of the pristine MWCNTs consists of two broad low-intensity bands, 531–534 eV and 540–544 eV, which, according to the XPS and NEXAFS data (Figure 2c,d), are due to the presence of oxides on the MWCNT outer surface.

A comparison of the O 1s absorption spectra of copper oxides and powder (Figure 10a) indicates that the main absorption bands of the latter correspond to a superposition of the bands of the CuO and Cu_2_O O 1s NEXAFS spectra. At the same time, an additional intense broad band at the photon energy of 538.5 eV is observed in the spectrum of the Cu/MWCNTs, which is absent in the CuO and Cu_2_O O 1s NEXAFS spectra. In addition, the O 1s absorption spectrum of the nanocomposite contains a low-intensity band at 531.4 eV characteristic of CuO. The presence of CuO is confirmed by the XPS Cu 2p spectra, indicating that the CuO is at least 76% of the copper compounds in the nanocomposite. However, the low intensity of this peak in the Cu/MWCNTs spectrum remains unclear. It can be shown by normalizing the O 1s NEXAFS partial spectra of the CuO and the nanocomposite to the first peak (531.4 eV) with the subsequent subtraction of the copper oxide contribution (Figure 10b) that the 538.5 eV band cannot be associated with the CuO presence in Cu/MWCNTs. Besides, the presence of carbon oxides in the nanocomposite in the form of C=O, C–O–C, and C–O bonds and CO_3_ groups also does not explain the appearance of this band since the π* and σ* resonances of these groups are located in the energy ranges of 531–534 eV and 540–544 eV, respectively [43,64,65,66,67]. As noted earlier, the CO_3_^2−^ anion is formed during the copper formate decomposition and is contained in the coating layer. The concentration of anions in the nanocomposite is comparable to other carbon oxides, which is observed in the C 1s XPS spectrum of Cu/MWCNTs in Figure 8b. These carbon oxides are contained both in the deposited nanoparticles and in the nanotube open surface.

The EDS studies of the uncoated nanocomposite surface showed (see the Appendix A) the doubling of the concentration of carbon oxides on such areas compared with the initial MWCNTs. Therefore, taking into account that only a part of the initial MWCNT surface is covered by copper compounds (Figure 5), its O 1s XPS and NEXAFS spectra can be used to estimate the relative O-atoms concentration included in these oxides. Figure 10a shows that the O 1s NEXAFS spectrum of the MWCNTs consists of two broad low-intensity bands, 531–534 eV and 540–544 eV. These bands cannot make a significant contribution to the absorption in the 538.5 eV band in the nanocomposite spectrum. According to the XPS Cu 2p spectral data, the copper atoms in the nanocomposite deposited nanoparticles are in a bivalent state in the CuO compound. However, the O 1s NEXAFS spectrum indicates the insignificant relative concentration of O-atoms included in CuO and carbon oxides. Thus, one can assume the presence of another bivalent copper compound in the nanocomposite. Since XPS can provide complementary information to soft X-ray NEXAFS data, let us consider the PE O 1s spectra in Figure 10c. It can be seen from the figure that, as in the case of absorption spectra, the powder PE O 1s spectrum is completely determined by the CuO and Cu_2_O spectra, characterized by a broad band with a BE of 530.8 eV, close to the Cu_2_O O 1s level BE at 530.2 eV [68]. However, a broad band (531.6 eV) is observed in the O 1s spectrum of the nanocomposite, shifted by 0.8 eV towards a higher BE as compared with the band in the spectrum of the powder. For this band, a four-component approximation was performed, and the contributions responsible for the C=O (531.6 eV), C–O, C–O–C (533.3 eV) bonds in functional groups and bonds in CO_3_ (530.8 eV) and CuO (529, 8 eV) were separated. From a comparison of the peak integral intensities, the relative contents of oxygen atoms in the carbon oxides, copper oxide CuO, and a compound corresponding to the 531.6 eV band in the nanocomposite were estimated as 27%, 11%, and 62%, respectively. As noted above, the main contribution to the oxygen spectra is made by the bivalent copper compounds. In addition to the formation of CuO oxide, a Cu–O–C bond is allowed. This bond ensures good adhesion of the copper coating to the nanotube surface. The oxygen peak depends strongly on the nature of the metal surface. This peak at 529–531 eV corresponds to 3d-metal oxides, but, for metal–O–C bonds, it shifts at 1–4 eV to the high BE [19,20,21,63,69]. In the case of copper, this shift is ~1.6 eV [20], which is in good agreement with the measured 1.8 eV shift of the Cu–O–C peak relative to the BE of the CuO peak shown in Figure 10b. This makes it possible to identify the band with a 531.6 eV BE in the nanocomposite O 1s XPS spectrum as belonging to the Cu–O–C group. Moreover, the 538.5 eV wide band in the O 1s NEXAFS spectrum should also be associated with this group.

## 4. Conclusions

Due to the correctly selected chemical composition of the copper-containing coating on the MWCNT outer surface with an insignificant presence of carbon oxides, it was possible to obtain a coating of nanosized CuO/Cu_2_O/Cu nanoparticles, most of which were semitransparent for X-ray photoelectrons. The use of a complementary set of research methods allowed, for the first time, to explicitly identify the bands in the O 1s NEXAFS and XPS spectra of the Cu/MWCNTs hybrid nanocomposite, confirming the formation of oxygen bridges between the CuO nanoparticles and MWCNT outer surface in the Cu/MWCNTs hybrid nanocomposite. The Cu–O–C binding through oxygen bridges demonstrated in this work could be extended to other metal/MWCNTs composites to achieve the optimal characteristics of the new materials, which is crucial for various applications. The obtained results allow further studies to be conducted to determine the optimal conditions for the formation of Cu–O–C bridge bonds during the deposition of copper coatings on the surface of graphitized and carbonized materials.

## Figures and Tables

**Figure 1 nanomaterials-11-02993-f001:**
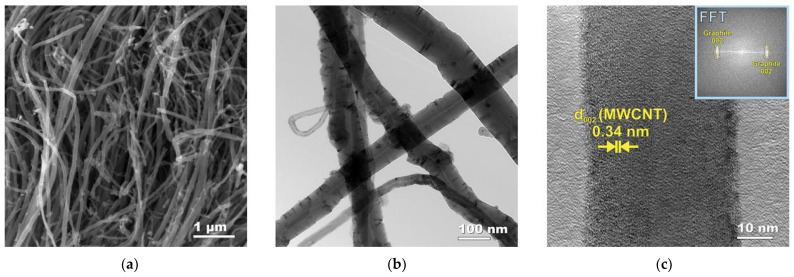
Electron microscopy images of the initial MWCNTs: (**a**) SEM; (**b**) TEM; (**c**) HRTEM of a lonely nanotube with the FFT pattern in insert.

**Figure 2 nanomaterials-11-02993-f002:**
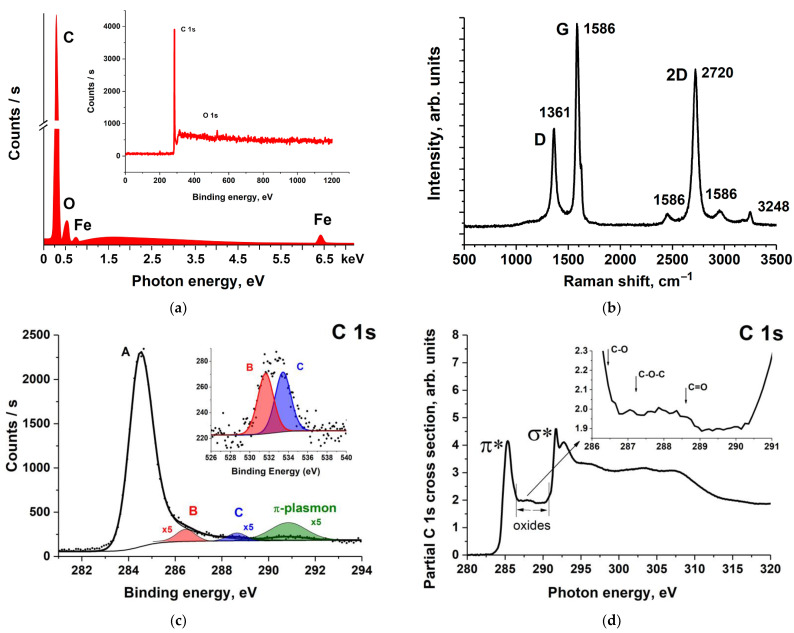
Characterization of the initial MWCNTs: (**a**) EDS (bottom) and XPS (top) survey spectra, (**b**) Raman spectra, (**c**) C 1s and O 1s XPS spectra, and (**d**) C 1s NEXAFS spectrum.

**Figure 3 nanomaterials-11-02993-f003:**
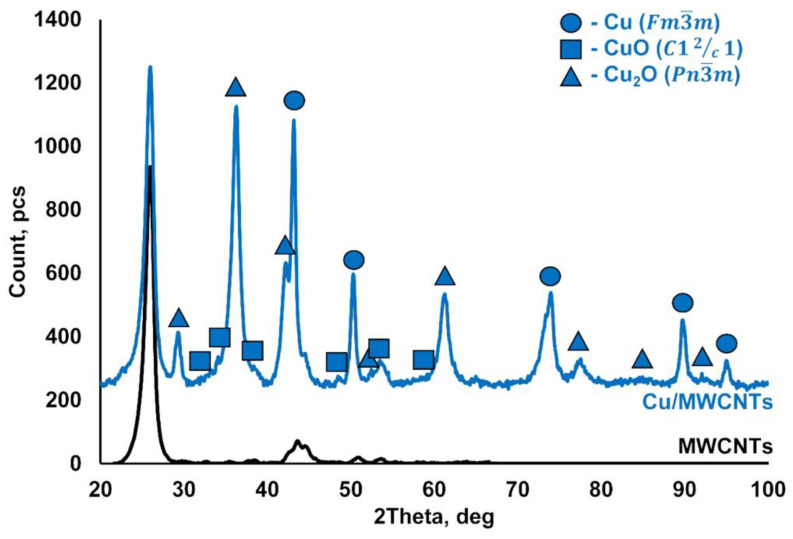
Powder XRD patterns of the initial MWCNTs (black) and Cu/MWCNTs hybrid nanocomposite (blue).

**Figure 4 nanomaterials-11-02993-f004:**
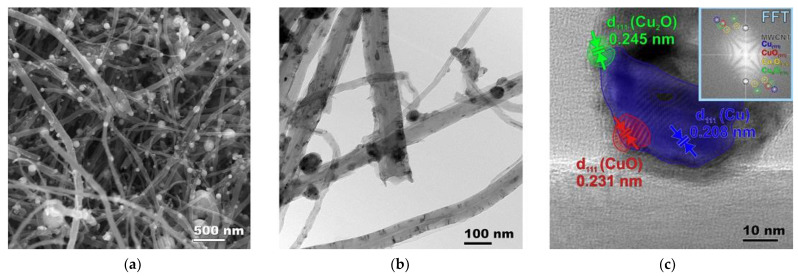
Electron microscopy images of the Cu/MWCNTs hybrid nanocomposite: (**a**) SEM; (**b**) TEM; (**c**) HRTEM of a lonely nanotube with the FFT pattern in insert.

**Figure 5 nanomaterials-11-02993-f005:**
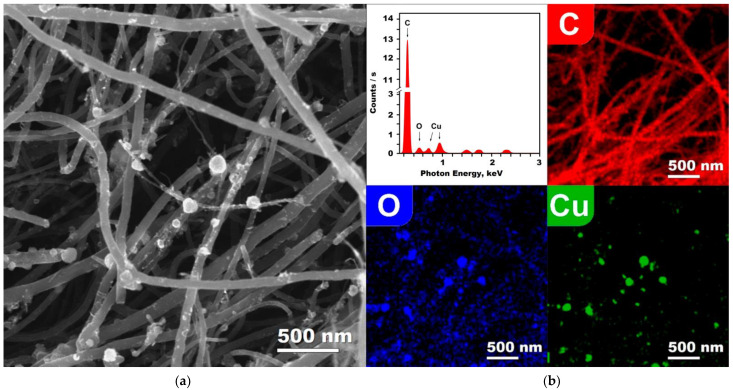
(**a**) SEM image and (**b**) EDS data of the Cu/MWCNTs hybrid nanocomposite: EDS spectrum and maps of spatial distribution of carbon, oxygen, and copper atoms.

**Figure 6 nanomaterials-11-02993-f006:**
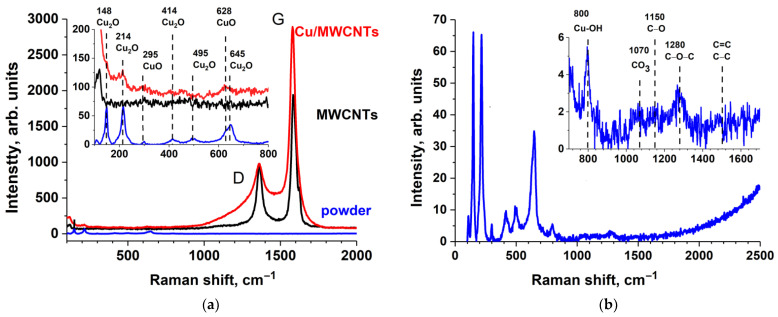
Raman spectra of (**a**) the initial MWCNTs, Cu/MWCNTs hybrid nanocomposite, and (**b**) powder from the reactor walls formed during the composite synthesis.

**Figure 7 nanomaterials-11-02993-f007:**
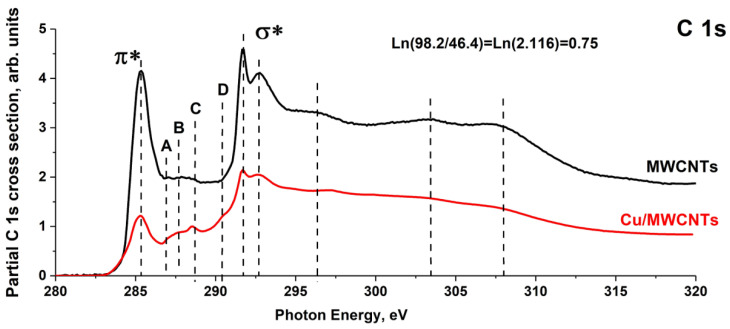
C 1s NEXAFS partial spectra of the initial MWCNTs and the Cu/MWCNTs hybrid nanocomposite.

**Figure 8 nanomaterials-11-02993-f008:**
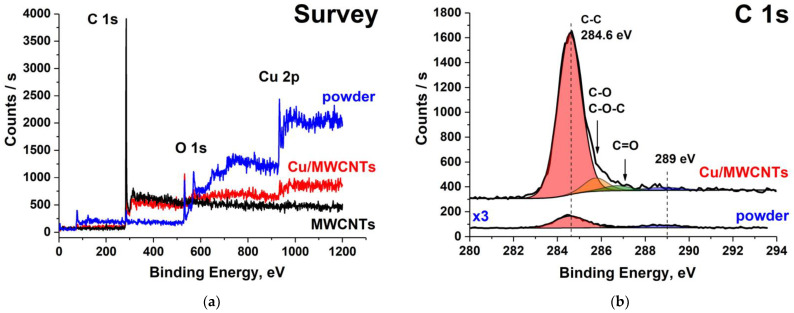
XPS spectra of the initial MWCNTs, Cu/MWCNTs hybrid nanocomposite, and powder from the reactor walls: (**a**) survey and (**b**) for the C 1s. The intensity of the powder C 1s spectrum was increased by 3 times.

**Figure 9 nanomaterials-11-02993-f009:**
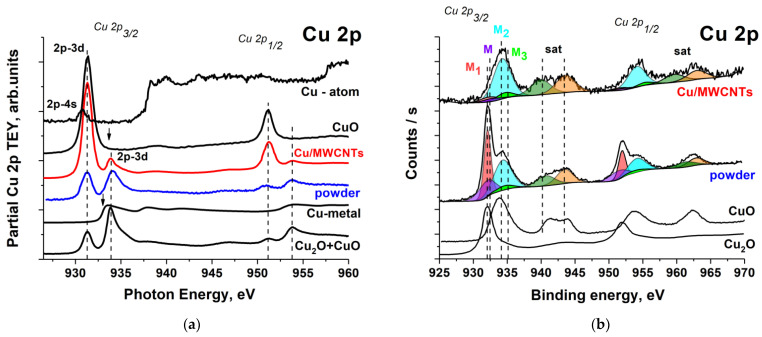
(**a**) comparison of the Cu 2p NEXAFS partial spectra of powder, Cu/MWCNTs hybrid nanocomposite, copper oxide CuO, cuprous oxide Cu_2_O with a low admixture of CuO, metallic copper, and photoion yield spectrum of atomic copper [55]. The arrows indicate the BE of the 2p_3/2_ electrons [56] in metallic copper (932.6 eV) and copper oxide CuO (933.6 eV). (**b**) Cu 2p XPS spectra of powder, copper oxide CuO, Cu/MWCNTs hybrid nanocomposite, and cuprous oxide Cu_2_O [57]. All the spectra were normalized to the TEY signal at the photon energy of 960 eV.

**Figure 10 nanomaterials-11-02993-f010:**
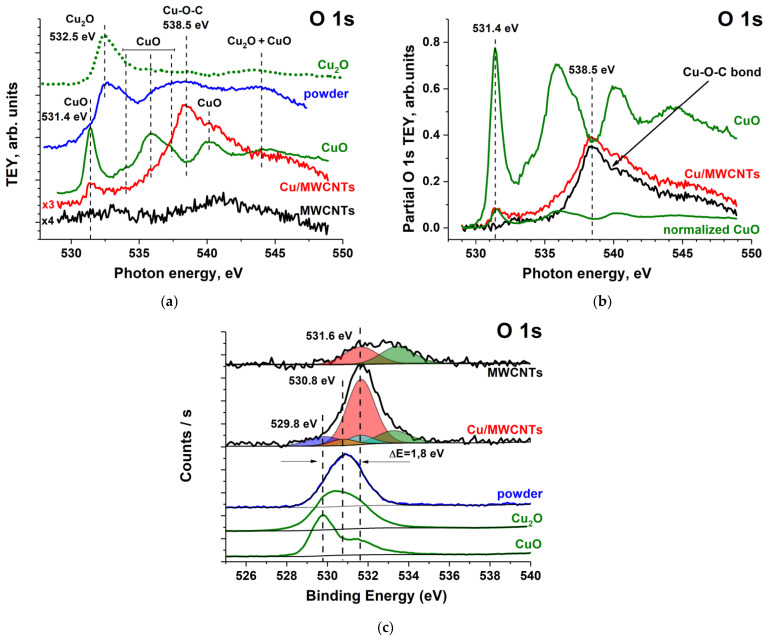
(**a**) The measured O 1s NEXAFS spectra of the initial nanotube, Cu/MWCNTs hybrid nanocomposite, powder and CuO, and Cu_2_O O 1s NEXAFS spectrum taken from [61]; (**b**) the TEY partial spectra in the O 1s NEXAFS region from Cu/MWCNTs hybrid nanocomposite, CuO oxide, CuO oxide normalized to the first peak (531.4) in the composite and Cu–O–C contribution is obtained as the difference between the nanocomposite and normalized CuO spectra; (**c**) The measured O 1s XPS spectra of the initial nanotube, Cu/MWCNTs hybrid nanocomposite and powder, and CuO and Cu_2_O O 1s XPS spectra taken from [57].

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
