# Peer review of "The Identification of Cu–O–C Bond in Cu/MWCNTs Hybrid Nanocomposite by XPS and NEXAFS Spectroscopy"

_nanomaterials, 2021, doi:10.3390/nano11112993_

Round 1

Reviewer 1 Report

This paper discusses Cu/MWCNT hybrid nanocomposite. Experimental,  measurement, evaluation, and discussion were comprehensively conducted. Therefore, the paper seems to be suitable for publication in Nanomaterials Journal. However, the reviewer thinks that additional discussion and information must be required for some points as the followings. When they will be cleared, the reviewer will recommend that the paper should be published.

1) In this study, the authors chose MWCNTs. What are permissible conditions? For example, diameters, length, structures, rates of a defect, and so on.

Or, are any MWCNTs permissible?

Also, how about single-walled CNTs?

2) Regarding Q1, the authors described the data of the prepared MWCNTs. But, only average diameter and length range were described. If the data can fit a normal distribution, for example, the authors must describe 1σ of dispersion.

3) In p. 7, the authors described the diameter of nanoparticles. As same as above, if the data can fit a normal distribution, the authors must describe 1σ of dispersion.

4) The NEXAFS measurements showed a peak caused by C=O group. The reviewer thinks this peak is caused by carboxy group (-COOH, it contains C=O coupling). Can C=O and COOH be distinguish by the NEXAFS?

If not, the authors should discuss for it.

Reviewer 2 Report

Review of

“The identification of Cu–O–C bond in Cu/MWCNTs hybrid nanocomposite by XPS and NEXAFS spectroscopy”

Contents

The manuscript deals with nanocomposites based on multi-walled carbon nanotubes (MWCNTs) decorated with CuO/Cu2O/Cu nanoparticles. Scanning and transmission electron microscopy, X-ray diffractometry and Raman and ultrasoft X-ray spectroscopy are applied in the research showing the formation of oxygen atom bridges between carbon atoms of the outer graphene layer and oxygen atoms of CuO and Cu2O oxides. The Cu–O–C bond is identified in the manuscript by using a complementary set of methods and the obtained results could be useful for further research in the field of nanocomposites.

Comments

The topic falls within the journal scope.

A revision is required to address the following issues.  

  • The adopted methodologies were previously exploited by the same research team in [14] with some rather expected and coincident outcomes. For instance, the conclusion on page 4 at lines 194-195: “The SEM studies revealed the presence of MWCNTs with different diameters (Figure 1 a)” was already contributed by the same authors in [14, Section 3.1]: “The SEM studies revealed the presence of MWCNTs with different diameters (Figure 1c)”.

Thus, merits and innovative implications (if any) should be illustrated and commented upon in comparison with previous contributions on the matter.

  • Characteristics of Cu/MWCNTs hybrid nanocomposites improved by Cu–O–C bonds should be better emphasized.

  • Advantages of using the metal organic chemical vapor deposition method should be clearly provided.

  • Reference values of mechanical strength, specific surface area and chemical inertia of multi-walled carbon nanotubes should be declared.

  • A clear illustration about the plan followed in the paper is missing and could be conveniently included in the introductory section.

Reviewer 3 Report

The present paper reported spectroscopic experiments of XPS and NEXAFS on Cu/MWCNTs hybrid composite. The main claim of the research, as described in the abstract, is the measurement of the Cu-O-C bonding peak that is important for decorating a MWCNT with Cu nanoparticles. The manuscript is written comprehensively with detailed characterizations of the samples and very fundamental information on the spectroscopy interpretations. In addition, the processing scheme is described in the supplementary. The manuscript is organized with a collection of the data and it seems to be appropriate for non-specialists for their understandings of the individual results. However, it may need the further effort to have consistency between the data. My comments are following:

1) Results on the sample start with Fig.3 and can be summarized as follows:

Figure 3: XRD, showing large signals from Cu and Cu2O. The triangle (Cu2O) and circle (Cu) XRD peaks are much larger than the square ones (CuO). The Cu2O size is 20 nm and the CuO size is 3nm.

Figure 4, 5: Microscopic images of MWCNT, covered by a number of the Cu (oxide) nanoparticles

Figure 6: Raman spectra, showing NO vibration modes of the Cu-O-C bonding

Figure 7: C 1s NEXAFS, showing negligible peak of the C-O bonds. 

Figure 8: C 1s XPS, showing only the negligible spectral tail that probably contain a peak of the C-O bond.

Figure 9: Cu 2p NEXAFS/XPS, showing apparent material peaks of CuO and copper bicarbonate.

Figure 10: O 1s NEXAFS/XPS, showing huge peaks of the Cu-O-C bond.    

The authors explained that signals of the Cu-O-C bond was hidden in the C 1s and Cu 2p spectra due to large signals of MWCNT and Cu nanoparticles, respectively. On the other hand, judging from the Figs. 4,5, the Cu-O-C bonds are formed at the interface between MWCNT and Cu nanoparticles that attach very partly on the MWCNT surface. Thus, the interface signal should be very little, as expected from Figs. 4 and 5, but it appears significantly large in Fig.10. The authors need the additional explanation.  

2) Related to 1), XRD results in Fig.3 indicates that Cu oxides are mainly composed of Cu2O. But the spectroscopic data in the paper show that contribution of CuO is much larger than that of Cu2O. The authors need the explanation in the paper.

3)Related to 1), the authors wrote as follows in page 7:

XRD) The averaged crystallite sizes calculated from line broadening of Cu (111), Cu2O (111) and CuO (020) reflections are about 40, 20 and 3 nm, respectively.

SEM) The diameter of nanoparticles ranges from 50 to 150 nm although the overwhelming majority of them have a size of about 100 nm. The thickness of the oxide shell is about 5 nm, and the sizes of copper oxide nanocrystallites are approximately of 2 – 3 nm.

   The quantitative arguments are confusing and seem not consistent to each other.

4) What is a reason to show spectra of powder from the reactor walls, Fig.6(b)? It is not necessary for discussion of the paper.

5) In Figs. 2, 8, 9 and 10, there were curve-fits in the spectra but the details were not described anywhere in the paper. The authors need to show the fitting functions, the Gaussian-width, the Lorentzian width, and the residue, for examples.

6) Related to 3), the fitting function is Fig.8 seems to be Gaussian rather than Voigt or Lorentzian that has been typically adopted for the high-resolution XPS. The authors claimed that the spectral tail in Fig.8 was the evidence of the C-O bonds but it might by just a part of the Voigt or Lorentzian peak of the C-C bond.

7) Labels, M, B, C, in Fig. 10 (c), should be used for the discussion in the main text in the page 15.

Round 2

Reviewer 1 Report

The reviewer can recommend this paper for publication.

Reviewer 2 Report

The paper is now suitable for publication in Nanomaterials.